# Quercetin and Its Mixture Increase the Stress Resistance of *Caenorhabditis elegans* to UV-B

**DOI:** 10.3390/ijerph17051572

**Published:** 2020-02-29

**Authors:** Shi-ming Li, Dan Liu, Yi-lin Liu, Bin Liu, Xing-huang Chen

**Affiliations:** 1College of Food Science, Fujian Agriculture and Forestry University, Fuzhou 350002, China; shiming.li.27sep@gmail.com (S.-m.L.); 1180940002@fafu.edu.cn (D.L.); lynn201806@sina.com (Y.-l.L.); 2National Engineering Research Center of JUNCAO Technology, Fujian Agriculture and Forestry University, Fuzhou 350002, China

**Keywords:** *Caenorhabditis elegans*, UV-B, quercetin, luteolin, lycopene, apoptosis, stress resistance

## Abstract

Ultraviolet B (UV-B, 280–320 nm) radiation causes complex molecular reactions in cells, including DNA damage, oxidative stress, and apoptosis. This study designed a mixture consisting of quercetin, luteolin and lycopene and used *Caenorhabditis elegans* as a model to study the resistance of these natural chemicals to UV-B. Specifically, we have confirmed that quercetin and its mixture can increase the resistance of *Caenorhabditis elegans* to UV-B through lifespan test, reactive oxygen species level assay, germ cell apoptosis test, embryonic lethal test and RT-qPCR experiments. The results show that quercetin and its mixture prolonged the lifespan of UV-B-irradiated *Caenorhabditis elegans* and reduced abnormal levels of reactive oxygen species, embryo death, and apoptosis induced by UV-B. The protective effect of quercetin and its mixture may be attributed to its down-regulation of *HUS-1*, *CEP-1*, *EGL-1* and *CED-13.* Therefore, the results of this research could help the development of UV-B radiation protection agents.

## 1. Introduction

Radiation is divided into two forms: ionizing radiation and non-ionizing radiation. Ultraviolet (UV) radiation is the most common non-ionizing radiation in human daily life. Common UV rays are divided into UV-C (200–280 nm), UV-B (280–320 nm) and UV-A (315–340 nm) according to wavelength. UV radiation as an effective mutagen, which has been proved by epidemiology and molecular biology to causes human skin cancer [1]. UV irradiation can cause DNA double helixes to deform, producing thymine dimers, cyclobutane pyrimidine dimers (CPD) and other pyrimidine dimers. These deformed DNA fragments are usually recognized and removed by the nucleotide excision repair (NER) system [2]. In addition, UV radiation also increases the level of reactive oxygen species (ROS) including hydroxide (OH^−^), superoxide anion (O2^−^), singlet oxygen (O_2_) and hydrogen peroxide (H_2_O_2_) in mitochondria [3]. High levels of ROS activate the mitogen-activated protein kinase pathway, which causes damage to DNA, leading to cell cycle arrest and apoptosis [4]. Therefore, UV radiation can directly or indirectly cause damage to DNA. Among them, UV-A and UV-B are radiation that people often receive, and UV-B has stronger carcinogenic effect than UV-A [5].

As early as 1949, researchers first reported in vivo studies of the protective effect of antioxidants on ionizing radiation, demonstrating that cysteine protects rats from lethal doses of X-rays [6]. Later, in 1985, researchers first reported that 2% beta-carotene and 0.07% canthaxanthin can effectively prevent the development of skin tumors caused by UV-B [7]. Subsequently, a large number of chemicals like flavonoids with anti-oxidant activity have been reported to inhibit radiation damage [8,9]. The development of natural compounds with low side effects to inhibit UV-induced damage has become a popular subject.

Generally, UV radiation activates death receptors and mitochondrial apoptotic pathways in cells, which triggers apoptosis [10]. The *CEP-1* in *C. elegans* is a homologous gene of *p53* in mammals, and their functions are almost the same, mainly mediating apoptosis and meiotic chromosome separation [11]. The HUS-1 is a conserved checkpoint protein. The classical apoptotic pathway mediated by CEP-1 requires HUS-1 to activate, so it is necessary for DNA damage-induced cell cycle arrest and apoptosis [12]. After activation of the CEP-1-mediated apoptotic pathway, transcription of the pro-apoptotic genes *EGL-1* and *CED-13* is activated. The EGL-1 triggers apoptosis induction by antagonizing the functions of anti-apoptotic members of the BCL-2 family [12]. In addition, CED-13 is the only protein with a BH3 domain in *C. elegans*, which interacts with CED-9 to promote apoptosis when overexpressed [13]. These four genes are involved in the regulation of UV-induced apoptosis in *C. elegans* (Figure 1).

The natural chemicals used in this experiment are quercetin, luteolin and lycopene. The molecular formula of quercetin (3, 3′, 4′, 5, 7-pentahydroxyflavone) is C_15_H_10_O_7_. It has strong antioxidant activity [9], antitumor activity [14], and radiation resistance [15]. Similarly, luteolin (3′, 4′, 5, 7-tetrahydroxyflavone, C_15_H_10_O_6_) and lycopene (C_40_H_56_) also have great antioxidant and antitumor effects [16,17]. Therefore, this study designed a mixture using these three natural chemicals. The effect and mechanism of quercetin and its mixture on increasing stress resistance of *C. elegans* to UV-B were studied by lifespan test, ROS level assay, germ cell apoptosis test, embryonic lethal test and RT-qPCR.

## 2. Materials and Methods

### 2.1. Strain, Cultivation and Grouping of C. elegans

The N2 wild-type Bristol strain *C. elegans* used in this experiment were purchased from *Caenorhabditis* Genetics Center (University of Minnesota, MN, USA). They are usually cultured in nematode growth medium (NGM) plates (6 cm in diameter) at 20 °C and given *E. coli* OP50 as food [18]. Before each test, *C. elegans* were synchronized with the lysate (1 M NaOH solution: 5% NaClO solution = 1:1) to ensure that they could grow to L4 state at the same time. The eggs obtained by the synchronization operation could grow to the L4 stage after being cultured at 20 °C for 2.5 days in NGM plates [19].

All *C. elegans* are administered at the L4 stage. A temporary mixed solution (100 μL OP50 solution + 100 μL drug solution) was added dropwise to the surface of each NGM plate. The natural extracts used in this study were quercetin (Q4951, Sigma, St. Louis, MO, USA), luteolin (L107329, Aladdin, Los Angeles, CA, USA) and lycopene (FY1235, Feiyubio, Nantong, China). When these substances were dissolved, DMSO (D8370, Solarbio, Beijing, China) needed to be added so that the final concentration of DMSO was 0.5%. In the pre-experiment of the effect of monomers and mixtures on the inhibition of UVB-induced damage, the *C. elegans* were grouped as follows: (the dose is given in parentheses) CT (0.5% DMSO), UV (0.5% DMSO), Q (1 μM quercetin), Lu (1 μM luteolin), Ly (1 μM lycopene), QLu (0.5 μM quercetin + 0.5 μM luteolin), QLy (0.5 μM quercetin + 0.5 μM lycopene), Mixture (MIX, 0.33 μM quercetin + 0.33 μM luteolin + 0.33 μM lycopene). After that, we designed the mixture composition. In the experiment of quercetin and its mixture inhibiting UVB-induced damage, the grouping of elegans was as follows: CT (0.5% DMSO), UV (0.5% DMSO), Quercetin Low-dose (QL, 0.5 μM quercetin), Quercetin High-dose (QL, 1 μM quercetin), Mixture Low-dose (ML, 0.16 μM quercetin + 0.16 μM luteolin + 0.16 μM lycopene), Mixture High-dose (MH, 0.33 μM quercetin + 0.33 μM luteolin + 0.33 μM lycopene). We ensured that the total molar value of the drugs in each administration group was the same (1 μM). The NGM plates were changed daily with fresh drugs and OP50 were added during the administration period to prevent the drug from invaliding in the plate due to prolonged oxidation.

### 2.2. UV-B Irradiation

A UV Cross-linker (JRA03-II, Jieruian, WuXi, China) that can produce UV-B (312 nm) was used to irradiate the *C. elegans* at a dose of 100 J/M^2^, which is a more suitable dose that causes damage to *C. elegans* cells [1]. The irradiation operation was a multi-lamp irradiation mode (5 lamps) of the UV cross-linker. A UV integrator (UV-Int150, UV-Design, ZheJiang, China) was used to calibrate the exposure dose. When irradiating *C. elegans*, the covers of the NGM plates were removed, and it was confirmed that there was no liquid on the surface of the culture medium to exclude the interference of the liquid medium on the radiation dose.

### 2.3. Photographing of C. elegans

When photographing the results of each test, more than 10 *C. elegans* were transferred to a microscope slide with a 3% agarose pad. Ten microliters of a 30 μmol/mL NaN_3_ solution was added dropwise to anesthetize *C. elegans* [20]. Photographs of FITC (fluorescein isothiocyanate), and BF (bright field) were taken using a fluorescence electron microscope (Scope.A1, Carl Zeiss Jena Inc., Baden-Württemberg, Germany) as needed.

### 2.4. ROS Level Assay

There were two ROS level tests in this study, and two experiments were performed according to the group mentioned above. Thirty *C. elegans* grown to L4 state after synchronization in each group were transferred to a new NGM plate, and raised with OP50 solution mixed with different drugs for 48 h. After transferring them to new NGM plates, they were irradiated with UV-B (normal light was used in the CT group) at a dose of 100 J/M^2^, and then fed with OP50 without drugs for 6 h. The *C. elegans* was placed in 1.5 mL centrifuge tubes and washed twice with M9 buffer (3 g KH_2_PO_4_, 6 g Na_2_HPO_4_, 5 g NaCl, 1 mL of 1 M MgSO_4_, and H_2_O to 1 L), and then transferred to 100 μL M9 buffer containing 10 μmol 2′,7′-dichlorofluorescein (A601220, Sangon, Shanghai, China) for staining in the dark (20 °C, 30 min) [4]. Ten *C. elegans* were randomly selected from each group, and the dichlorofluorescein (DCF) was photographed using a fluorescence microscope (50X, FITC) as described above. Finally, Image J software (National Institutes of Health, Maryland) was used to analyze the fluorescence intensity of head muscle cells of *C. elegans*.

### 2.5. Lifespan Test

The synchronized L4 state *C. elegans* were transferred to NGM plates fed with different drugs according to groups. There was a total of 10 groups, and each group was repeated with 2 plates, with 25 worms on each plate [21]. Divided into UV treatment group (UV, QL-UV, QH-UV, ML-UV, MH-UV) and without UV treatment group (CT, QL, QH, ML, MH), a total of 500 worms were tested. The dosage was as described above. All groups were fed at 20 °C for 48 h, recorded as day 0, and they were no longer administered. On day 0, all *C. elegans* were transferred to new drug-free NGM plates and fed at 20 °C, and the UV-treated groups were irradiated with a dose of 100 J/M^2^ UV-B. From the first day, the number of alive and dead worms was recorded daily and the NGM medium was changed daily to prevent adult individuals from confounding with offspring individuals. The standard for the dead worms was no movement and swallowing, and no response after touching.

### 2.6. Gonadal Cell Apoptosis Test

The synchronized adult *C. elegans* were fed for 48 h using the same method of administration as described above, with 30 worms in each group. After transferring them to new NGM plates, they were irradiated with UV-B (normal light was used in the CT group) at a dose of 100 J/M^2^, and then fed with OP50 without drug for 6 h. They were washed twice with M9 buffer in 1.5 mL centrifuge tubes, and then transferred to 100 μL M9 buffer containing 1 mmol of Acridine orange (AO, A8120, Solarbio, Beijing, China) for staining in the dark (20 °C, 1 h). After that, they were transferred into new NGM plates for 1 h to let the worms excrete AO dye in the digestive tract. According to the above photographing method, the gonad arm cells of the worms were photographed under a fluorescence microscope 100× field of view (FITC) [22]. The apoptotic cells of each gonad arm were counted by the double-blind method.

### 2.7. Embryo Lethality Test

The synchronized L4 stage *C. elegans* were transferred to NGM plates with different drugs for 48 h. They were irradiated with UV-B at a dose of 100 J/M^2^ (normal light in the CT group). Six randomly selected individuals from each group were placed on blank NGM plates to lay eggs, with two worms per NGM plate, and three replicate plates per group. After allowing them to lay eggs at 25 °C for 4 h, they were transferred to new NGM plates and the number of eggs was counted. Subsequently, this operation was repeated twice. These *C. elegans* laid eggs for a total of 12 h at 25 °C. After 24 h at 25 ° C, non-hatched eggs (dead eggs) were counted on all NGM plates [23].

### 2.8. RT-qPCR Analysis of Transcription Levels of HUS-1, CEP-1, EGL-1, and CED-13

The synchronized L4 stage *C. elegans* were transferred to NGM plates with different drugs for 4 days, with 40 worms in each group. After transferring them to new NGM plates, they were irradiated with UV-B (normal light was used in the CT group) at a dose of 100 J/M^2^, and then fed with OP50 without drug for 6 h. After that, each group was transferred to a 1.5 mL centrifuge tube containing M9 buffer and washed 5 times to remove contaminants, respectively. Their RNA was extracted according to the instructions of the Ultrapure RNA Kit (CW0597, Cowin Bio, Beijing, China). According to the instructions, the obtained RNA was reverse transcribed into cDNA using PrimeScript ™ RT reagent Kit (RR047A, TaKaRa, Dalian, China), and quantitative PCR analysis was performed. The first strand of cDNA was synthesized in a multifunctional PCR instrument (ABI/ProFlex, Thermo Fisher Scientific, Waltham, MA, USA). Quantitative PCR was performed in the ABI 7300 PCR system (Thermo Fisher Scientific, Waltham, MA, USA) using TB Green Premix Ex Taq II with ROX (RR820A, TaKaRa, Dalian, China). Each PCR reaction system contained 20 uL, which consisted of 10 μL of TB Green Premix solution, 8 μM forward primer, 8 μM reverse primer, 0.4 μL Rox Dye and 2 μL cDNA. The reactions were performed in 200 μL PCR 8-Tube (F600552, BBI, Shanghai, China), and these tubes were loaded into 96 wells of the PCR system. The PCR cycle conditions were as follows: 1 cycle of 30 s at 95 °C for denaturation, and then 40 cycles of 5 s at 95 °C and 31 s at 60 °C for the PCR reaction. The melting curves were checked at the end of the cycle to confirm the specificity of the primers in each reaction. The PCR reaction was repeated three times. The results were calculated by the 2^−ΔΔ*C*t^ method. *ACT-1* was used as a reference gene [24]. All primers used are shown in Table 1.

### 2.9. Statistical Analysis

All data in this paper were calculated and statistically analyzed (Student’s *t*-test and Log-rank test) using Microsoft Excel 2007 (Microsoft Corp., Washington, DC, USA) and GraphPad 7 (GraphPad Inc., San Diego, CA, USA) software packages.

## 3. Results

### 3.1. Initial Evaluation of Different Drugs on the Resistance to UV-B Stress by ROS Assay

In this experiment, the effects of different drugs on the resistance to UV-B stress were evaluated by the ROS level test in *C. elegans*, and the composition of quercetin mixture was designed based on the results. In the results of mean fluorescence intensity (MFI) of ROS (Figure 2), the MFI of UV group was significantly higher than that of other groups (** *p* < 0.01, Student’s *t*-test). The ROS level of MIX group (0.33 μM quercetin + 0.33 μM luteolin + 0.33 μM lycopene) decreased significantly and was lower than that of other drug treatment-groups (^##^
*p* < 0.01, Student’s *t*-test). There were no significant differences between the mean fluorescence intensities of the other drug-treated groups. Therefore, in the subsequent experiments, we decided to use the composition of the MIX group as the mixture composition.

### 3.2. Quercetin and Its Mixture Helped C. elegans Resist UV-B Stress

Figure 3a shows the effect of quercetin and its mixture on the lifespan of *C. elegans* under UV-B stress. Among them, the CT group, UV group, and MH-UV group had a maximum survival duration of 24 days, 15 days, and 20 days, respectively. Table 2 shows the median survival duration of each group. The median survival duration of the CT group, UV group, and MH-UV group were 10.5 days, 4.5 days, and 7 days. The drug-treated group significantly prolonged worms after exposure to UV-B (Log-rank test), of which the MH-UV group performed the best (*** *p* < 0.001). Figure 3b shows the effect of quercetin and its mixture on the lifespan of normal *C. elegans*, which proves that the 48-h administration has no significant effect on the lifespan test (*p* > 0.05). These results indicate that quercetin and its mixture increase the resistance of *C. elegans* to UV-B stress.

### 3.3. Quercetin and Its Mixture Inhibit UV-B-Induced Increase in ROS Levels

Figure 4 shows the inhibition effects of quercetin and its mixture at different dose on increased ROS levels caused by UV-B in *C. elegans*. UV-B dramatically increased ROS levels in worms. The levels of ROS in all drug treatment groups were significantly lower than those in the UV group (** *p* < 0.01, Student’s *t*-test), and the effect of the mixture high-dose (MH) was the most significant (^##^
*p* < 0.01, Student’s *t*-test). In addition, Figure 3 shows bright-field photos of the same area of FITC.

### 3.4. Quercetin and Its Mixture Decreased DNA Damage of Germ Cells Caused by UV-B

Six hours after the *C. elegans* were exposed to UV-B radiation (100 J/M^2^), the apoptosis results (Figure 5) of the gonad arm show that UV-B caused DNA damage to germ cells, thereby increasing the level of apoptosis. The level of germ cell apoptosis in quercetin and its mixture group was significantly lower than that in the UV group (** *p* < 0.01). Similarly, in the results of dead embryo counts (Figure 6), the embryo lethality in the QH, ML, and MH groups was significantly lower than that in the UV group (** *p* < 0.01), and that of MH group was significantly lower than other drug-treatment groups (^#^
*p* < 0.05). Therefore, quercetin and its mixture reduced DNA damage of gonad arm cells caused by UV-B in *C. elegans*.

### 3.5. Regulation of Quercetin and Its Mixture on Apoptosis Related Genes in C. elegans

Figure 7 shows that UV-B caused an increase in the expression level of apoptosis-related genes (*HUS-1*, *CEP-1*, *EGL-1* and *CED-13)* in *C. elegans*. Except for the QL group, the drug treatment groups significantly reduced the expression levels of these genes (* *p* < 0.05, ** *p* < 0.01). The MH group had the most significant regulatory effect on HUS-1, CEP-1 and EGL-1 (^#^
*p* < 0.05, ^##^
*p* < 0.01). In the expression results of CED-13, the expression levels of MH and QH groups were both significantly lower than those of other treatment groups. This result indicates that quercetin and its mixture inhibited UV-B induced apoptosis in *C. elegans*.

## 4. Discussion

In this study, we used ROS level assays to screen and design a mixture with UV-B resistance in *C. elegans* at first. After that, we verified that quercetin and its mixture can increase the resistance of *C. elegans* to UV-B stress by lifespan tests, ROS level tests, germ cell damage assessments, and RT-qPCR experiments of apoptosis-related genes. Among them, the high-dose mixture (0.33 μM quercetin + 0.33 μM luteolin + 0.33 μM lycopene) has the most significant resistance to cell damage caused by UV-B in *C. elegans*.

UV irradiation can cause a variety of cellular responses, including cell damage and apoptosis. The most critical mechanisms are the accumulation of reactive oxygen species and the mitochondrial pathway of apoptosis [25]. The large accumulation of ROS promotes oxidative stress and apoptosis, resulting in cellular DNA damage [26]. Therefore, in this study, the protective effect of quercetin, luteolin and lycopene on UV-B-induced damage was evaluated by using a convenient and rapid ROS level assay, and the composition of the quercetin mixture was designed based on the results. In subsequent results, quercetin and its mixture have been shown to significantly inhibit the increase in ROS levels caused by UV-B, similar to the results of previous studies by others [27]. In the lifespan experiment after UV-B irradiation, quercetin and its mixture significantly prolonged the maximum survival days and median survival days of *C. elegans* under UV-B stress. In addition, a 48-h drug treatment control experiment showed that this procedure did not significantly affect the lifespan of *C. elegans* without UV-B irradiation. These results indicate that quercetin and its mixture significantly improved the stress resistance of *C. elegans*, which is similar to the conclusions of others [28]. As the results show, the reduction in abnormal ROS levels and the increase in lifespan after UV irradiation means that quercetin and its mixture can reduce the level of oxidative stress. This may be one of the mechanisms that increase the resistance of *C. elegans* to UV-B stress.

Apoptosis is the process of programmed cell death controlled by intrinsic genes in order to maintain homeostasis under certain physiological conditions [29]. UV causes apoptosis through a complex process, including directly destroying DNA, increasing p53 levels, and directly or indirectly activating cell death receptors [30]. In the results of germ cell damage evaluation, quercetin and its mixture significantly reduced UV-B induced DNA damage, so the number of apoptotic cells in the gonad arms and dead embryos of the drug-treated group was significantly lower than that of the UV group. It shows that quercetin and its mixture reduce the DNA damage of *C. elegans* caused by UV-B by some mechanisms, thereby increasing the resistance stress of worms. Later, in RT-qPCR results of apoptosis-related genes, we discovered one of these mechanisms. After DNA is damaged by UV-B, the expression levels of pro-apoptotic genes *EGL-1* and *CED-13* are up-regulated [31]. During this classic apoptosis activation process, the mechanism of BH3 domain proteins in *C. elegans* is similar to that of mammals [12]. Therefore, the apoptosis response of *EGL-1* and *CED-13* corresponding DNA damage must be mediated by *CEP-1* [32], and the activation of *CEP-1* also requires the checkpoint protein HUS-1 [31]. The results of RT-qPCR show that quercetin and the mixture in the high-dose group could significantly inhibit the increase in expression levels of *HUS-1*, *CEP-1*, *EGL-1* and *CED-13* caused by UV-B, which explained their mechanism of inhibiting apoptosis [33]. However, the effects of quercetin on *HUS-1*, *CEP-1* and *CED-13* in the low-dose group were not significant.

Overall, this study found that a mixture of quercetin, luteolin, and lycopene can increase the resistance of *C. elegans* to UV-B stress. According to our findings in these experiments, these substances may increase the resistance of *C. elegans* by two mechanisms, including the inhibition of UV-B-induced abnormal oxidative stress and abnormal apoptosis. This is of great significance for the development of protective agents that increase UV resistance by using *C. elegans* as the animal model in the future. In addition, more detailed UV-resistant mechanisms of quercetin, luteolin and lycopene should be researched, and whether there is a biological synergy among them. Most importantly, future studies should be better performed using mice and monkeys to discover the protective mechanism of antioxidant drugs against UV-induced damage in mammals.

## Figures and Tables

**Figure 1 ijerph-17-01572-f001:**
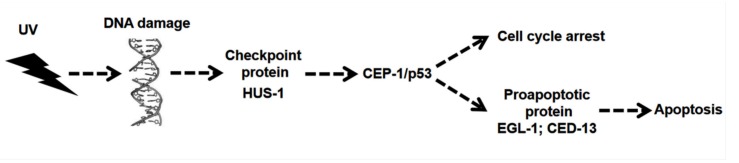
UV-induced apoptosis pathway in *C. elegans*.

**Figure 2 ijerph-17-01572-f002:**
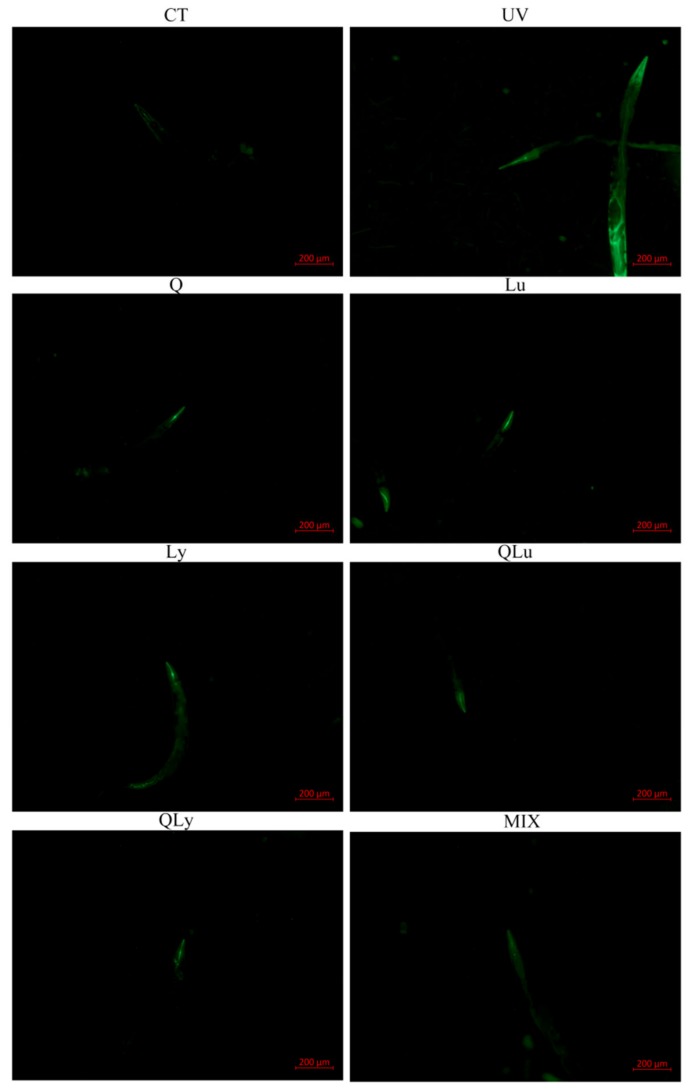
The mean fluorescence intensity (MFI) results of *C.elegans* stained by 2′,7′-dichlorofluorescein show the reactive oxygen species (ROS) levels of every group. These pictures were taken in a 50× field of view (FITC). The higher the ROS level, the brighter the green fluorescence. All data are mean ± SEM and were analyzed by two-tailed Student’s *t*-test, *n* = 7, * *p* < 0.05, ** *p* < 0.01 compared to UV group. ^#^
*p* < 0.05, ^##^
*p* < 0.01 compared to MIX group.

**Figure 3 ijerph-17-01572-f003:**
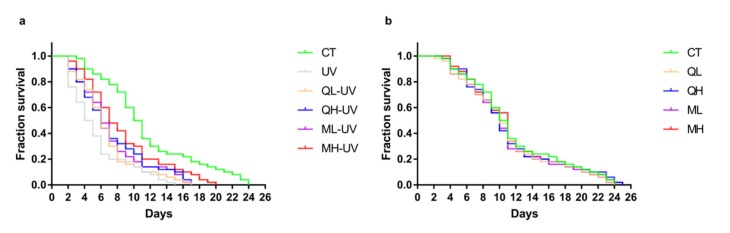
Lifespan test results of *C. elegans*. (**a**) The survival curves of the drugs + UV (100 J/M^2^) treatment groups. (**b**) The survival curves of the drug treatment groups without UV irradiation (*n* = 50).

**Figure 4 ijerph-17-01572-f004:**
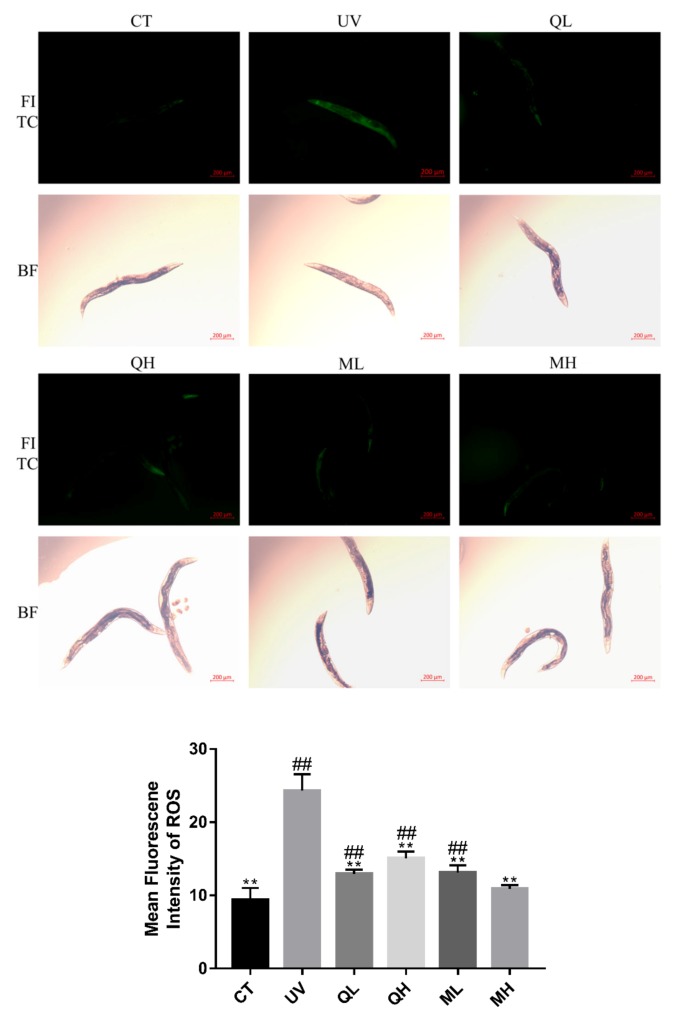
The MFI results of *C.elegans* stained by 2′,7′-dichlorofluorescein show the ROS levels of every groups. These pictures were taken in a 50× field of view (FITC and BF). The higher the ROS level, the brighter the green fluorescence. All data are mean ± SEM and were analyzed by two-tailed Student’s *t*-test, *n* = 7, * *p* < 0.05, ** *p* < 0.01 compared to UV group. ^#^
*p* < 0.05, ^##^
*p* < 0.01 compared to MH group.

**Figure 5 ijerph-17-01572-f005:**
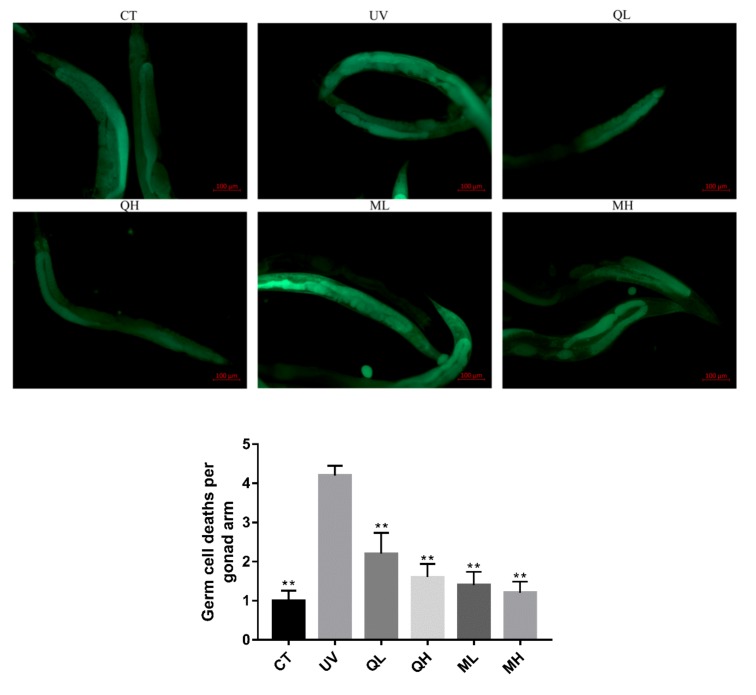
Apoptosis results of gonad arm cells of every group 6 h after UV-B irradiation at a dose of 100 J/M^2^. These pictures were taken in a 100× field of view (FITC). The tiny yellow-green foci in the picture are the apoptotic cell stained by AO. All data are mean ± SEM and were analyzed by two-tailed Student’s *t*-test, *n* = 10, * *p* < 0.05, ** *p* < 0.01 compared to UV group.

**Figure 6 ijerph-17-01572-f006:**
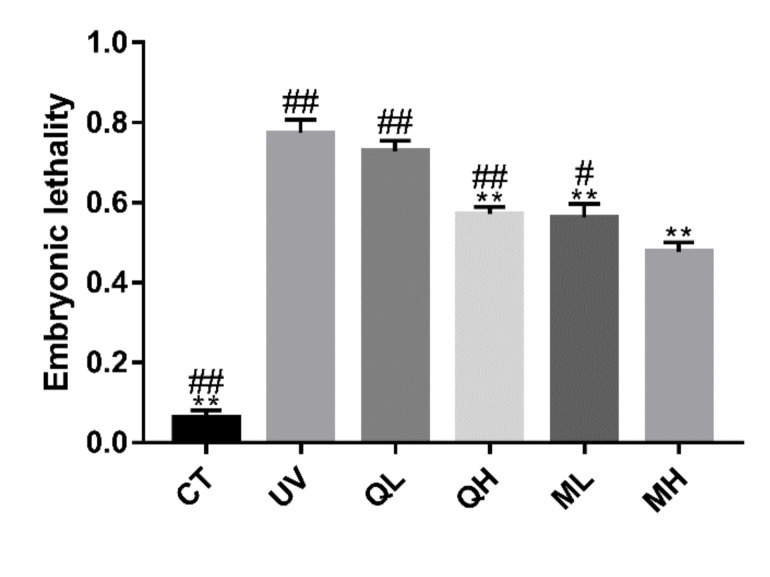
Results of embryonic lethality of every group after UV-B irradiation (100 J/M^2^). All data are mean ± SEM and were analyzed by two-tailed Student’s *t*-test, *n* = 6, * *p* < 0.05, ** *p* < 0.01 compared to UV group. ^#^
*p* < 0.05, ^##^
*p* < 0.01 compared to MH group.

**Figure 7 ijerph-17-01572-f007:**
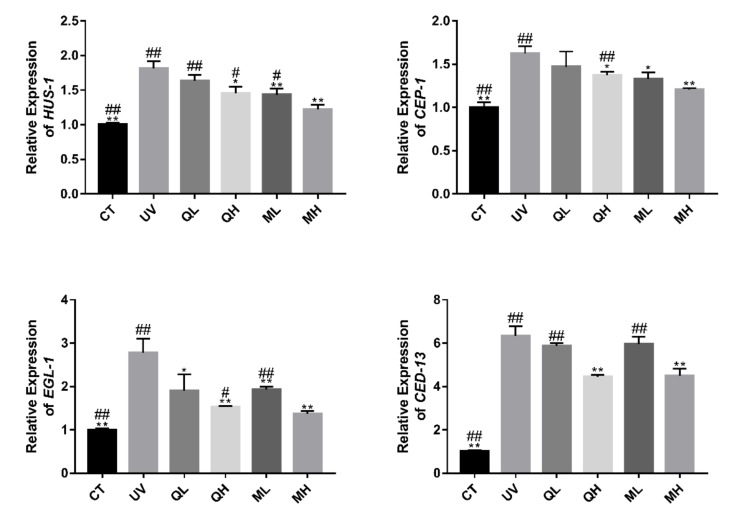
Changes in the relative expression of apoptosis-related genes *HUS-1*, *CEP-1*, *EGL-1* and *CED-13*. All data are mean ± SEM and were analyzed by two-tailed Student’s *t*-test, *n* = 3, * *p* < 0.05, ** *p* < 0.01 compared to UV group. ^#^
*p* < 0.05, ^##^
*p* < 0.01 compared to MH group.

**Table 1 ijerph-17-01572-t001:** Primer sequence listing of PCR.

Genes	Forward (5′-3′)	Reverse (5′-3′)
*ACT-1*	GCTCTTGCCCCATCAACCAT	AGAAAGCTGGTGGTGACGAT
*HUS-1*	GGCAATCGACGTGTTTATCAAAAT	TCGTTTCGTGGATTCATGCC
*CEP-1*	TGTCCAGAAAATGATAGACGGAGT	GCATCGGAAATCTTTGGCGT
*EGL-1*	ACACCCAAAACATTCACACCG	GGCAAAGGTGAGCATCAGCA
*CED-13*	TCGAGGGCAGAAAAACGTGA	ACAACAGCGGGAGAAAGTGT

**Table 2 ijerph-17-01572-t002:** Analysis of survival curves for each UVB-treated group.

Groups	Median Survival (d)	*p* Value (Log-Rank)
CT	10.5 ****	<0.0001
UV	4.5	-
QL-UV	6 *	0.0770
QH-UV	6 *	0.0169
ML-UV	6 *	0.0149
MH-UV	7 ***	0.0003

* Compared with UV group. The significance analysis was calculated by the Log-rank (Mantel-COX) test. * *p* < 0.05, *** *p* < 0.001, **** *p* < 0.0001.

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
