# Peer review of "Quercetin and Its Mixture Increase the Stress Resistance of Caenorhabditis elegans to UV-B"

_ijerph, 2020, doi:10.3390/ijerph17051572_

Round 1

Reviewer 1 Report

Authors focused on the effects of quercetin, luteolin and lycopene on resistance of C. elegans to UV-B stress. Various methods were used including the life-span test, ROS level measurement, germ cell damage and RT-PCR tests. Pure substances or combinations of substances in different concentrations were tested. The mixture of quercetin, luteolin and lycopene (at highest doses) was the most effective in improving the stress resistance.

The text is written briefly and clearly, the same is true for figures and pictures.

Recommendations:

  1. A mixture of quercetin, luteolin and lycopene was tested. However, authors write about „a compound“. The world „ compound“ evokes a pure chemical substance, not a mixture. This terminology is misleading, and should be changed. Chemical formulas of quercetin, luteolin and lycopene can be added to the article. Or authors suppose a chemical reaction between quercetin, luteolin and lycopene and a new compound formation? It should be clearly explained.
  2. In Introduction part, the apoptotic pathways which are addressed in the study should be described with details. A schematic figure showing the roles of CEP-1, HUS-1, EGL-1, CED-13 could be added.
  3. Important issue: a multiple control for each substance (or mixture) tested should be done – e.g. for quercetin: control (only DMSO), UV (stress), quercetin, quercetin+UV. All these groups should be defined, compared and discussed. I suppose these experiments were done but there are not clearly reported. E.g. did the substances (or mixtures) applied without UV stress produce any effects?
  4. In Discussion, the results could be interpreted with details, and especially the possible mechanism(s) of protection against UV-B stress deserve(s) more attention.
  5. Experiments with worms are interesting and can bring new data. It would be interesting to obtain data also from other animal studies. Authors should at least propose the direction of future research.

Author Response

Reviewer 1:

Authors focused on the effects of quercetin, luteolin and lycopene on resistance of C. elegans to UV-B stress. Various methods were used including the life-span test, ROS level measurement, germ cell damage and RT-PCR tests. Pure substances or combinations of substances in different concentrations were tested. The mixture of quercetin, luteolin and lycopene (at highest doses) was the most effective in improving the stress resistance.

The text is written briefly and clearly, the same is true for figures and pictures.

Recommendations:

  1. A mixture of quercetin, luteolin and lycopene was tested. However, authors write about „a compound“. The world „ compound“ evokes a pure chemical substance, not a mixture. This terminology is misleading, and should be changed. Chemical formulas of quercetin, luteolin and lycopene can be added to the article. Or authors suppose a chemical reaction between quercetin, luteolin and lycopene and a new compound formation? It should be clearly explained.

- This description error has been fixed. "Compound" in the manuscript was changed to "mixture".

  1. In Introduction part, the apoptotic pathways which are addressed in the study should be described with details. A schematic figure showing the roles of CEP-1, HUS-1, EGL-1, CED-13 could be added.

- The more detailed description to the apoptotic pathways of these four genes is in the discussion section. A schematic of this apoptotic pathway has been added in the introduction.

  1. Important issue: a multiple control for each substance (or mixture) tested should be done – e.g. for quercetin: control (only DMSO), UV (stress), quercetin, quercetin+UV. All these groups should be defined, compared and discussed. I suppose these experiments were done but there are not clearly reported. E.g. did the substances (or mixtures) applied without UV stress produce any effects?

- As the result of the lifespan test in the manuscript, the lifespan of the drug-treated C.elegans without UV stress was almost the same as that of the CT group. There was no effect on the mixture fed by C. elegans without UV stress. So we don't think it is necessary to perform the experimental group without UV stress in other experiments.

  1. In Discussion, the results could be interpreted with details, and especially the possible mechanism(s) of protection against UV-B stress deserve(s) more attention.

- The more detailed discussion has been added.

  1. Experiments with worms are interesting and can bring new data. It would be interesting to obtain data also from other animal studies. Authors should at least propose the direction of future research.

- Future research directions have been added.

Reviewer 2 Report

The study of the effect of anti-oxidants on the UV irradiation resistance is an important topic which deserves thorough investigation. The presented research is interesting and the paper should be published. I have noticed several issues that need correction.

  1. Provide complete list of abbreviations at the beginning of the paper. 
  2. line 44: "....number of chemical elements like flavonoids..." flavonoids are compounds, not elements.
  3. When describing the effect of the "compound" (as a chemist, I don't agree with the word, because this is actually a mixture of compounds), is is not clear what the contribution of each component is. The authors can make a great point if they study the effect of each ingredient separately, and then the combination of the compounds. This is actually mentioned in the procedure, however it is not clear if each of the components showed any effect, and if so, how is this comparable with the "compound'.
  4. line 173: " Initial evaluation of the resistance of different drugs to UV-B stress by ROS assay"... this is actually the effect of the drugs on the resistance. The title contradicts the content of the paragraph.
  5. There are no captions under the photographs. Maybe they have to be presented as separate figures.

Author Response

The study of the effect of anti-oxidants on the UV irradiation resistance is an important topic which deserves thorough investigation. The presented research is interesting and the paper should be published. I have noticed several issues that need correction.

  1. Provide complete list of abbreviations at the beginning of the paper. 

-A list of abbreviations has been added.

  1. line 44: "....number of chemical elements like flavonoids..." flavonoids are compounds, not elements.

- This error has been modified.

  1. When describing the effect of the "compound" (as a chemist, I don't agree with the word, because this is actually a mixture of compounds), is is not clear what the contribution of each component is. The authors can make a great point if they study the effect of each ingredient separately, and then the combination of the compounds. This is actually mentioned in the procedure, however it is not clear if each of the components showed any effect, and if so, how is this comparable with the "compound'.

- This is indeed a misleading description. The original "compound" in the text and the figure has been replaced by "mixture".

  1. line 173: " Initial evaluation of the resistance of different drugs to UV-B stress by ROS assay"... this is actually the effect of the drugs on the resistance. The title contradicts the content of the paragraph.

- This is a description error and has been fixed.

  1. There are no captions under the photographs. Maybe they have to be presented as separate figures.

- The photo in the manuscript shares a caption with its corresponding chart, which includes a description of the content of the photo. They are described together because the chart is a quantification of the photo.

Round 2

Reviewer 1 Report

Authors have provided a new version of their article titled “Quercetin and its compound increase the stress resistance of Caenorhabditis elegans to UV-B” where they reflected the referees' comments. I do not have any other recommendations.